

# Influence of aprepitant on the pharmacodynamics and pharmacokinetics of gliclazide in rats and rabbits

Raghunandan Reddy Kura[1,*], Eswar Kumar Kilari[1,*] and Mastan Shaik[2]

[1] Pharmacology Division, Andhra University College of Pharmaceutical Sciences, Andhra University, Visakhapatnam, Andhra Pradesh, India
[2] Medical Services, Troikaa Pharmaceuticals Ltd., Ahmedabad, Gujarat, India
[*] These authors contributed equally to this work.

Corresponding author
Mastan Shaik, shkmastan@gmail.com, mastanshaik@troikaapharma.com

## ABSTRACT

**Background**. Concomitant drug administration is a general phenomenon in patients with chronic diseases such as diabetes mellitus. Among the currently available oral antidiabetic drugs, gliclazide is a commonly prescribed drug considering its multiple benefits in diabetic patients. Aprepitant is a commonly prescribed antiemetic drug which is mainly metabolized by CYP3A4, reported to have modest inductive and inhibitory effects on CYP2C9 and CYP3A4, respectively. Since gliclazide is metabolized by CYP2C9 (major) and CYP3A4 (minor), it is very difficult to predict the influence of aprepitant and its metabolic interaction with gliclazide. Considering the complexity associated with the combination of aprepitant and gliclazide, this study was designed to evaluate the influence of aprepitant on the pharmacodynamics (PD) and pharmacokinetics (PK) of gliclazide in animal models.

**Methods**. The PD interaction studies were conducted in both rodent (normal and alloxan-induced diabetic rats) and non-rodent (rabbits) animal models ($n = 6$) while the PK interaction study was conducted in normal rabbits ($n = 6$). An extrapolated human therapeutic oral dose of gliclazide, aprepitant and their combination were administered to rats and rabbits with 7 days washout between each treatment. For the multiple-dose interaction study, the same groups were administered with an interacting drug (aprepitant) for 7 days and then the combination of aprepitant and gliclazide on the 8th day. From the collected animal blood samples, blood glucose (by Glucose-Oxidase/Peroxidase method), insulin (by ELISA method) and gliclazide concentration levels (by HPLC method) were determined. Non-compartmental PK analysis was conducted by Phoenix WinNonlin software to determine the PK parameters of gliclazide. Statistical analysis was performed by student's paired $t$-test.

**Results**. The pharmacodynamic activity (blood glucose reduction and insulin levels) of gliclazide was significantly ($p < 0.05$) influenced by aprepitant in normal and diabetic condition without any convulsions in animals. There was a significant ($p < 0.05$) increase in concentration levels and Area Under the Curve of gliclazide while significant ($p < 0.05$) decrease in clearance levels of gliclazide in rabbits. The PK interaction with gliclazide is relatively more with the multiple dose treatment of aprepitant over single dose treatment.

**Conclusion**. In combination, aprepitant significantly influenced the pharmacodynamic activity of gliclazide in animal models. Considering this, care should be taken when this combination is prescribed for the clinical benefit in diabetic patients.

## INTRODUCTION

Concomitant drug administration is a general phenomenon in the patients with chronic diseases such as diabetes mellitus. Despite the several benefits with concomitant drug administration, it is generally associated with increased risks of medication non-adherence, adverse drug reactions/events, and toxicity due to potential drug interactions (*Peron, Ogbonna & Donohoe, 2015*). Monitoring and minimizing the risk of drug–drug interactions (either pharmacodynamic or pharmacokinetic) is a common goal in drug therapy while dealing with chronic diseases (*Curtis, 2006*). Diabetes mellitus is among the most prevalent and morbid chronic diseases which can adversely influence the health of millions of population worldwide (*Julie & John, 2017*).

The prevalence of diabetes mellitus has been dramatically increased from 2005 to 2015 by 30.6% (*Global Burden of Disease , GBD*). In high risk individuals, the overall prevalence of pre- and post-operative nausea and vomiting is in the range of 30 to 80% (*Bergese et al., 2012*). Among the currently available oral antidiabetic drugs, gliclazide is a commonly prescribed drug considering its multiple benefits (relatively low incidence of hypoglycemia, antioxidant activity and cardiovascular benefits) in diabetic patients (*Fava et al., 2002*; *O'Brien et al., 2000*; *Schernthaner, 2003*; *Ziegler & Drouin, 1994*). Aprepitant is a potent, selective and brain-penetrant non-peptide neurokinin-1 (NK1)-receptor antagonist, widely used for the prevention of chemotherapy induced nausea and vomiting (*Anna et al., 2017*). Aprepitant is mainly metabolized by CYP3A4 and reported to have modest inductive and inhibitory effects on CYP2C9 and CYP3A4, respectively (*Anna et al., 2017*; *Shadle et al., 2004*).

There is a high propensity for the concomitant use of aprepitant and gliclazide in diabetic patients due to (a) high prevalence of diabetes mellitus and nausea/vomiting, (b) the recent trend to recommend aprepitant for the management of refractory diabetic gastroparesis in diabetic patients (*Chong & Dhatariya, 2009*; *Fountoulakis et al., 2017*). Since gliclazide is metabolized (*Mastan & Kumar, 2009*; *Satyanarayana & Kilari, 2006*) by CYP2C9 (major) and CYP3A4 (minor), it is very difficult to predict the influence of aprepitant and its metabolic interaction with gliclazide. Considering the complexity associated with the combination of aprepitant and gliclazide, this study was designed to evaluate the influence of aprepitant on the pharmacodynamics (PD) and pharmacokinetics (PK) of gliclazide in rodent (normal and alloxan-induced diabetic rats) and non-rodent (rabbits) animal models.

## MATERIAL AND METHODS

### Materials

Aprepitant and gliclazide were gift samples from Dr. Reddy's Laboratories (Hyderabad, India). Glucose Kits (Span Diagnostics, Venette, France) were procured from a local

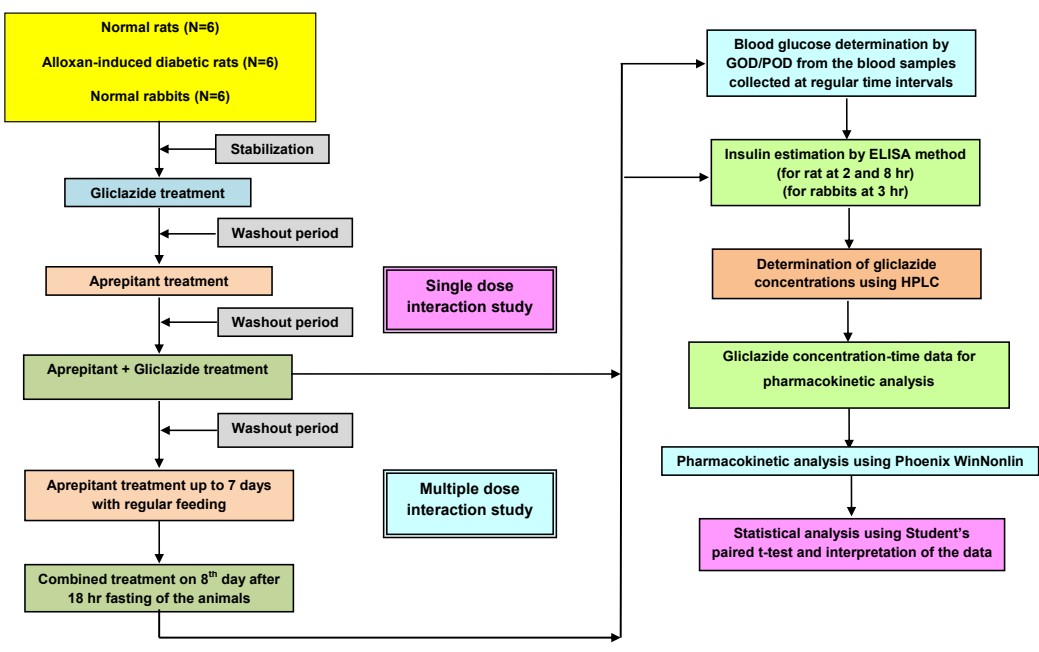

**Figure 1  Experimental design.**

pharmacy. Alloxan monohydrate, orthophosphoric acid, acetonitrile, dichloromethane and other relevant analytical reagents were procured from SD Fine Chemicals Limited, (Mumbai, India) and Loba Chemie Private Limited (Mumbai, India).

## Animals

The animals (albino rats and rabbits—both male and female) were purchased from Mahaveer Enterprises (Hyderabad, India). For this study, the rats with 220–270 g weight and 6-8 weeks of age, and rabbits with 1.35–1.75 kg weight and 3 months of age were selected. Throughout the study, animals were housed individually in polypropylene cages with specific identification number. The experimental animals were provided with a standard autoclaved commercial diet and filtered water *ad libitum*. Animals fasted for 18 h prior to the dosing and dietary restrictions were followed during experimentation.

## Study design

As represented in Fig. 1, this study was designed considering the all important factors to correlate the preclinical results to the clinical scenario. This study plan was approved by an Institutional Animal Ethics Committee (IAEC) (Registration Number 516/01/A/CPCSEA). The study was performed as per the recommendations of the Committee for the Purpose of Control, and Supervision of Experiments on Animals (CPCSEA) guidelines.

The main steps in this study are:

(1)  Selection of the animal models
(2)  Dose selection of the experimental drugs (aprepitant and gliclazide)
(3)  Pharmacodynamic interaction studies in normal and diabetic rats

(4)   Pharmacodynamic and pharmacokinetic interaction studies in rabbits
(5)   Pharmacokinetic and statistical analysis

In rats and rabbits, as represented in Fig. 1, the activity (PD and PK) of gliclazide was evaluated in the presence of acute (single dose administration) and successively chronic (multiple dose administration for 7 days) treatment with an interacting drug (aprepitant).

## Selection of the animal models

It is well known fact that the preclinical PK/PD results from one animal model cannot be correlated directly with humans due to the existence of interspecies differences. The identification and characterization of any interaction (PD or PK) in two dissimilar species at different conditions (normal and diabetic) will certainly validate the possible existence of that interaction in humans. Hence, the selection of relevant animal models is a crucial factor in the evaluation of drug–drug interactions. Rodents (normal and alloxan-induced diabetic rats) and non-rodents (rabbits) were selected as animal models ($n = 6$) for this study.

## Dose selection of the experimental drugs (aprepitant and gliclazide)

To mimic the clinical scenario, the dose of the experimental drugs was determined by extrapolating the human oral therapeutic dose (aprepitant = 125 mg; gliclazide = 80 mg) (Summary of Product Characteristics) to rats and rabbits based on body surface area. The human equivalent dose of aprepitant in rats and rabbits is 11.25 mg/kg b wt and 8.75 mg/1.5 kg b wt, respectively. The human equivalent dose of gliclazide in rabbits is 5.6 mg/1.5 kg b wt. Based on previous studies and considering the preliminary dose effect-relationship results (Fig. 2), the dose of gliclazide in rats was determined as 2 mg/kg b wt. Both gliclazide and aprepitant were administered to the respective animal groups by oral gavage (*Mastan & Kumar, 2009*).

## Pharmacodynamic interaction studies in normal and diabetic rats

Pharmacodynamic interaction studies were conducted in two groups (normal and alloxan-induced diabetic rats) containing six animals each. Diabetes in albino rats was induced by the intraperitoneal injection of alloxan monohydrate (100 & 50 mg/kg b wt) for 2 consecutive days prior to entry into the study. Rats with a blood glucose level >11.10 mmol/L 3 days following the injection were considered as diabetic and selected for the study (*Mastan & Kumar, 2009*).

For a single dose interaction phase, in each group, after an acclimatization period of 5 days, an oral dose of gliclazide was administered to each group. After a 7 day washout period, an oral single dose of interacting drug (aprepitant) was administered. After the 7 day washout period, a combination of single dose of aprepitant and gliclazide were administered with a time interval of 30 min. After this single dose interaction phase, the respective animal groups were continued for chronic treatment (multiple dose administration) with an interacting drug (aprepitant) for 7 days with a regular feeding. On the 8th day, a combination of single dose of aprepitant and gliclazide were administered with a time interval of 30 min.

Blood samples were collected at 0 (predose), 1, 2, 3, 4, 6, 8, 10 and 12 hr from retro orbital plexus of each rat. Blood glucose was determined from these samples by GOD/POD

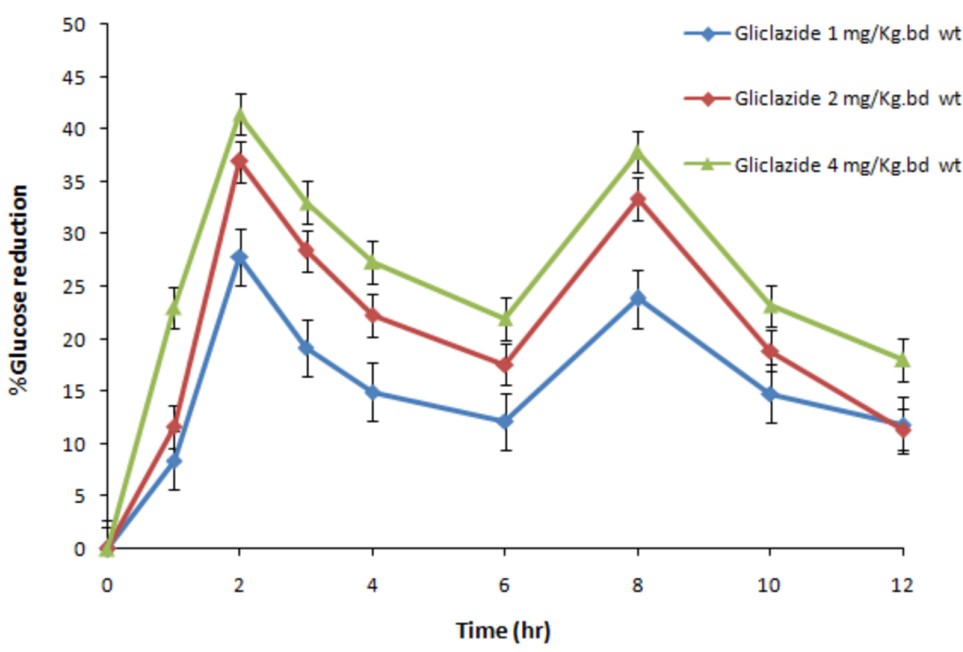

**Figure 2** **The dose-effect relationship of gliclazide on blood glucose in normal rats ($N = 6$).** Blue, Gliclazide 1 mg/kg bd wt.; red, Gliclazide 2 mg/kg bd wt.; green, Gliclazide 4 mg/kg bd wt.

method and serum insulin was determined (at 2 and 8 hr) by ELISA method (*Lequin, 2005*) using a commercial kit (Mercodia, Uppsala, Sweden) at Biological E Limited (Hyderabad, India).

## Pharmacodynamic and pharmacokinetic interaction studies in rabbits

The experimental design for in rabbits is similar to the rats as detailed above. The major difference in rabbits study is simultaneous determination of pharmacodynamic and pharmacokinetic parameters from the blood samples which were collected at 0 (predose), 1, 2, 3, 4, 6, 8, 10, 12, 16, 20 and 24 hr from the marginal ear vein of each rabbit. Blood glucose was determined by GOD/POD method and serum insulin was determined (at 3 hr) by ELISA method.

## Pharmacokinetic and statistical analysis

Gliclazide concentration levels were determined by HPLC method (*Kumar et al., 2007*; *Kumar & Mastan, 2011*). Non-compartmental PK analysis was conducted by Phoenix WinNonlin software to determine the PK parameters of gliclazide. The maximum concentration ($C_{max}$) and time for the maximum concentration ($T_{max}$) were observed values directly from the concentration and time data of gliclazide in rabbits. The Area Under the Curve (AUC) and Area Under the Moment Curve (AUMC) were determined by the linear trapezoidal rule. The elimination rate constant ($K_{el}$) was determined by log-linear regression of concentration–time data during the terminal elimination phase. The Mean Residence Time (MRT) was determined by the ratio of AUMC and AUC. The terminal

**Table 1** Mean percent blood glucose reduction of gliclazide in the presence and absence of aprepitant administration in normal and diabetic rats ($n = 6$).

| Time (hr) | Normal rats | | | | Alloxan induced diabetic rats | | | |
|---|---|---|---|---|---|---|---|---|
| | Gliclazide[#] | Aprepitant[^] | Aprepitant + Gliclazide (Single dose treatment)[a] | Aprepitant + Gliclazide (Multiple dose treatment)[b] | Gliclazide[#] | Aprepitant[^] | Aprepitant + Gliclazide (Single dose treatment)[a] | Aprepitant + Gliclazide (Multiple dose treatment)[b] |
| 1 | $10.30 \pm 3.42$ | $03.31 \pm 9.70$ | $21.01 \pm 6.54^{**}$ | $29.18 \pm 5.87^{***}$ | $13.20 \pm 1.71$ | $07.11 \pm 2.09$ | $20.34 \pm 1.39^{***}$ | $34.31 \pm 1.52^{***}$ |
| 2 | $36.99 \pm 2.90$ | $18.50 \pm 7.72$ | $52.29 \pm 6.62^{***}$ | $60.81 \pm 8.15^{***}$ | $39.83 \pm 1.54$ | $19.91 \pm 1.82$ | $56.36 \pm 1.87^{***}$ | $63.44 \pm 3.65^{***}$ |
| 3 | $30.14 \pm 5.59$ | $15.84 \pm 8.53$ | $44.42 \pm 7.46^{**}$ | $52.67 \pm 6.45^{***}$ | $30.73 \pm 2.75$ | $15.11 \pm 2.19$ | $39.15 \pm 2.70^{***}$ | $45.14 \pm 2.25^{***}$ |
| 4 | $26.42 \pm 8.73$ | $14.34 \pm 8.75$ | $41.85 \pm 5.53^{**}$ | $50.11 \pm 4.93^{***}$ | $28.52 \pm 2.22$ | $12.16 \pm 2.83$ | $36.91 \pm 3.20^{***}$ | $38.77 \pm 3.30^{***}$ |
| 6 | $19.51 \pm 9.90$ | $12.78 \pm 8.83$ | $35.17 \pm 5.71^{**}$ | $50.11 \pm 5.31^{***}$ | $21.82 \pm 1.20$ | $10.74 \pm 1.98$ | $33.48 \pm 2.84^{***}$ | $42.08 \pm 2.84^{***}$ |
| 8 | $35.94 \pm 2.66$ | $11.23 \pm 8.65$ | $49.42 \pm 8.63^{**}$ | $55.44 \pm 5.57^{***}$ | $38.91 \pm 3.68$ | $10.70 \pm 2.00$ | $51.30 \pm 2.61^{***}$ | $56.64 \pm 2.56^{***}$ |
| 10 | $19.64 \pm 8.29$ | $08.63 \pm 7.36$ | $26.46 \pm 6.97$ | $37.27 \pm 6.12^{**}$ | $31.68 \pm 2.98$ | $09.29 \pm 2.77$ | $36.49 \pm 2.30^{**}$ | $39.92 \pm 2.35^{***}$ |
| 12 | $10.40 \pm 12.89$ | $06.90 \pm 8.02$ | $20.75 \pm 5.96$ | $31.00 \pm 5.27^{**}$ | $16.42 \pm 1.85$ | $06.31 \pm 2.86$ | $26.12 \pm 2.06^{***}$ | $29.51 \pm 2.07^{***}$ |

**Notes.**

Data is represented as mean ± SD.

[#]2 mg/kg bd wt.

[^]11.25 mg/kg bd wt.

[a]Single dose treatment of aprepitant and single dose treatment of gliclazide.

[b]Multiple dose treatment of aprepitant (7 days) and single dose treatment of gliclazide.

[**]$P < 0.01$ compared to gliclazide group.

[***]$P < 0.001$ compared to gliclazide group.

half-life of gliclazide ($t_{1/2}$) was determined by the ratio of 0.693 and $K_{el}$ (*Kumar & Mastan, 2011*). Statistical analysis was conducted by student's paired $t$-test.

# RESULTS

## Pharmacodynamic interaction studies in rats

### Influence of aprepitant on gliclazide blood glucose reduction

The mean percent blood glucose reduction of gliclazide in presence and absence of aprepitant in normal and diabetic rats is represented in Table 1. In normal rats, gliclazide produced hypoglycemic activity with maximum biphasic glucose reduction of 36.99 ± 2.90% and 35.94 ± 2.66% at 2 and 8 hr. In diabetic rats, gliclazide produced anti-hyperglycemic activity with maximum biphasic glucose reduction of 39.83 ± 1.54% and 38.91 ± 3.68% at 2 and 8 hr. These results suggest that the blood glucose reduction with gliclazide treatment was higher in diabetic condition compared to normal condition. The maximum blood glucose reduction (18.50 ± 7.72% in normal rats and 19.91 ± 1.82% in diabetic rats) was observed at 2 hr with the interacting drug (aprepitant) in rats. The gliclazide blood glucose reduction was significantly increased in the presence of aprepitant in normal (single vs. multiple dose treatment: at 2 hr, 52.29 ± 6.62% and 60.81 ± 8.15%; at 8 hr, 49.42 ± 8.63% and 55.44 ± 5.57%) rats (Table 1). The gliclazide blood glucose reduction was significantly increased in the presence of aprepitant in diabetic (single vs. multiple dose treatment: at 2 hr, 56.36 ± 1.87% and 63.44 ± 3.65%; at 8 hr, 51.30 ± 2.61% and 56.64 ± 2.56%) rats (Table 1).

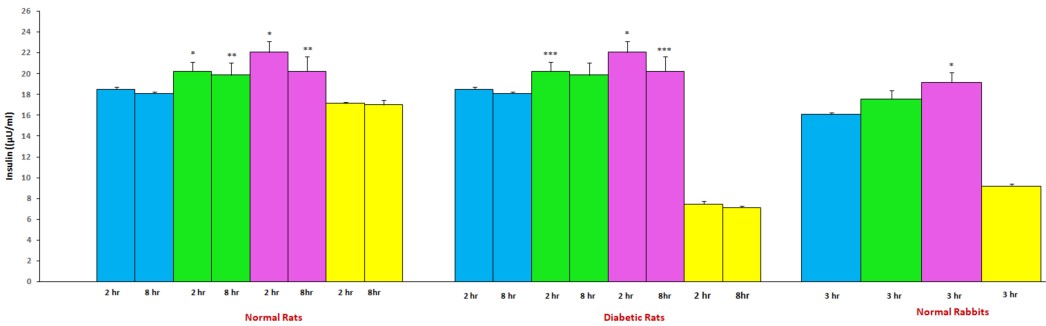

**Figure 3** **Mean insulin levels of gliclazide treatment in the presence and absence of aprepitant in rats and rabbits ($n = 6$).** Blue, Gliclazide treatment; green, single dose of aprepitant + Gliclazide; pink, multiple dose of aprepitant + Gliclazide; yellow, Aprepitant treatment. $*P < 0.001$ compared to gliclazide group; $**P < 0.01$ compared to gliclazide group; $*P < 0.05$ compared to gliclazide group;

## Influence of aprepitant on gliclazide insulin levels

The mean insulin levels of gliclazide in presence and absence of aprepitant in normal and diabetic rats is represented in Fig. 3. The insulin levels observed with gliclazide treatment were $18.48 \pm 0.24$ µU/mL (at 2 hr) and $18.08 \pm 0.18$ µU/mL (at 8 hr) in normal rats. In diabetic rats, insulin levels observed with gliclazide treatment were $9.33 \pm 1.43$ µU/mL (at 2 hr) and $8.19 \pm 1.24$ µU/mL (at 8 hr). In normal rats, the insulin levels observed with aprepitant treatment were $17.12 \pm 0.11$ µU/mL and $17.0 \pm 0.45$ µU/mL at 2 and 8 hr, respectively. In diabetic rats, the insulin levels observed with aprepitant treatment were $7.44 \pm 0.31$ and $7.14 \pm 0.10$ at 2 and 8 hr, respectively. The insulin levels were significantly increased in the presence of aprepitant in normal (single vs. multiple dose treatment: at 2 hr, $20.22 \pm 0.87$ and $22.05 \pm 1.05$ µU/mL; at 8 hr, $19.85 \pm 1.17$ and $20.24 \pm 1.39$ µU/mL) rats (Fig. 3). The insulin levels were significantly increased in the presence of aprepitant in diabetic (single vs. multiple dose treatment: at 2 hr, $11.25 \pm 1.15$ and $12.97 \pm 0.51$ µU/mL; at 8 hr, $9.09 \pm 0.83$ and $10.14 \pm 0.90$ µU/mL) rats (Fig. 3).

## Pharmacodynamic and pharmacokinetic interaction studies in rabbits

### Influence of aprepitant on gliclazide blood glucose reduction

The pharmacodynamic interaction results summary of gliclazide and aprepitant in rabbits is represented in Table 2. In rabbits, gliclazide produced hypoglycemic activity with maximum glucose reduction of $36.65 \pm 5.61\%$ at 3 hr. The blood glucose reduction observed with aprepitant was $14.14 \pm 3.11\%$ in rabbits. The gliclazide blood glucose reduction was significantly increased in the presence of aprepitant in rabbits (single vs. multiple dose treatment: $44.62 \pm 4.98\%$ and $48.87 \pm 5.17\%$).

### Influence of aprepitant on gliclazide insulin levels

The mean insulin levels of gliclazide in presence and absence of aprepitant in rabbits is represented in Fig. 3. The insulin level observed with gliclazide treatment was $13.19 \pm 0.90$ µU/mL at 3 hr in rabbits. The insulin level observed in rabbits with aprepitant treatment was $10.55 \pm 0.25$ µU/mL at 3 hr. The insulin levels were significantly increased

**Table 2  Mean percent blood glucose reduction of gliclazide in the presence and absence of aprepitant administration in normal rabbits ($n = 6$).**

| Time (hr) | Gliclazide[#] | Aprepitant[^] | Aprepitant + Gliclazide (Single dose treatment)[a] | Aprepitant + Gliclazide (Multiple dose treatment)[b] |
|---|---|---|---|---|
| 1  | 10.45 ± 3.41 | 02.19 ± 1.12 | 08.28 ± 2.07 | 09.67 ± 1.79 |
| 2  | 19.28 ± 6.69 | 05.40 ± 1.29 | 24.08 ± 4.76 | 26.13 ± 6.55 |
| 3  | 36.65 ± 5.61 | 09.73 ± 2.50 | 44.62 ± 4.98[*] | 48.87 ± 5.17[*] |
| 4  | 23.39 ± 5.12 | 14.14 ± 3.11 | 37.89 ± 4.17[***] | 41.18 ± 2.44[***] |
| 6  | 17.84 ± 4.52 | 12.19 ± 2.24 | 29.47 ± 3.41[***] | 34.37 ± 5.07[***] |
| 8  | 12.76 ± 5.80 | 09.79 ± 2.51 | 25.12 ± 4.17[**] | 30.75 ± 5.50[***] |
| 10 | 10.93 ± 5.39 | 08.13 ± 3.03 | 22.74 ± 4.52[**] | 26.93 ± 5.04[***] |
| 12 | 07.63 ± 5.66 | 06.23 ± 2.33 | 17.31 ± 4.52[**] | 23.52 ± 5.38[***] |
| 16 | 05.86 ± 4.91 | 04.38 ± 2.36 | 12.06 ± 4.80 | 19.27 ± 5.64[**] |
| 20 | 04.30 ± 4.51 | 02.67 ± 1.98 | 08.52 ± 5.21 | 14.84 ± 7.14[*] |
| 24 | 02.83 ± 4.07 | 01.54 ± 1.83 | 04.22 ± 5.33 | 11.53 ± 6.03[*] |

**Notes.**
Data is represented as mean ± SD.
[#] 5.6 mg/1.5 kg b.wt.
[^] 8.75 mg/1.5 kg bd wt.
[a] Single dose treatment of aprepitant and single dose treatment of gliclazide.
[b] Multiple dose treatment of aprepitant (7 days) and single dose treatment of gliclazide.
[*] $P < 0.05$ compared to gliclazide group.
[**] $P < 0.01$ compared to gliclazide group.
[***] $P < 0.001$ compared to gliclazide group.

in the presence of aprepitant in rabbits (single vs. multiple dose treatment: 14.16 ± 0.62 and 15.99 ± 0.79 μU/mL).

### Influence of aprepitant on gliclazide pharmacokinetics

The mean concentration and time profile of gliclazide in presence and absence of aprepitant is represented in Fig. 4. The influence of aprepitant on mean PK parameters of gliclazide is represented in Table 3. There was a significant increase in concentration levels and an alteration in the pharmacokinetic parameters (increase in $C_{max}$, AUC and MRT; decrease in clearance fraction) of gliclazide in the presence of aprepitant in rabbits.

## DISCUSSION

It is well known fact that several drugs have been withdrawn from the market due to their potent drug–drug interactions (*Thorir et al., 2003*). Hence, an evaluation of possible mechanisms of drug–drug interactions have become a cornerstone in the effective treatment of chronic diseases in which concomitant administration of multiple drugs is a very common phenomenon.

Considering the interspecies differences (*Alvares & Meyer, 1971*), this study was carefully designed to get precise and validated results from the selected animal models. The major advantages of this study are (i) an extrapolated human oral therapeutic doses of the experimental drugs to rats and rabbits to mimic the actual clinical scenario, (ii) it determined the pharmacodynamic activity (blood glucose reduction in specified time

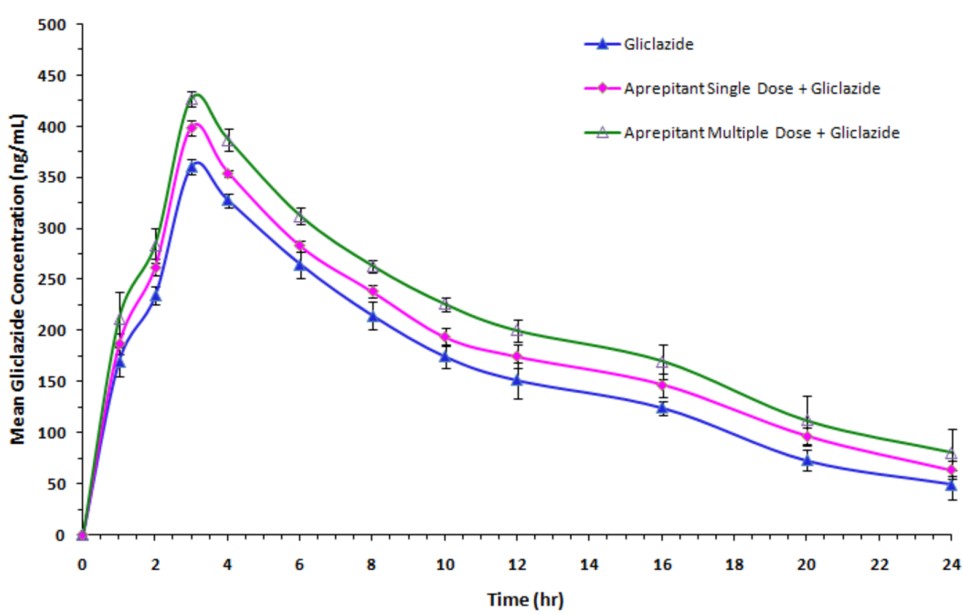

**Figure 4** **Mean serum gliclazide concentration-time profile of gliclazide in the presence and absence of aprepitant in rabbits ($n = 6$).** Blue, Gliclazide treatment; pink, single dose of aprepitant + Gliclazide; green, multiple dose of aprepitant + Gliclazide.

**Table 3** **Mean pharmacokinetic parameters of gliclazide before and after aprepitant administration in normal rabbits ($n = 6$).**

| Parameter | Gliclazide[#] | Aprepitant + Gliclazide (Single dose treatment)[a] | Aprepitant + Gliclazide (Multiple dose treatment)[b] |
|---|---|---|---|
| $C_{max}$ (ng/mL) | 360.11 ± 7.36 | 398.73 ± 7.42[**] | 426.97 ± 7.62[**] |
| $T_{max}$ (hr) | 03.0 ± 0.00 | 03.0 ± 0.00 | 03.0 ± 0.00 |
| CL (mL/hr) | 0.00127 ± 0.00 | 0.00110 ± 0.00[*] | 0.00094 ± 0.00[**] |
| Vd (mL) | 0.01284 ± 0.00 | 0.01202 ± 0.00 | 0.01114 ± 0.00 |
| $T_{1/2}$ (hr) | 7.12 ± 1.98 | 7.63 ± 1.38 | 8.49 ± 2.92 |
| $AUC_{0-24}$ (ng hr/mL) | 3,882.70 ± 125.54 | 4,407.63 ± 38.71[**] | 4,968.08 ± 155.17[**] |
| $AUC_{0-inf}$ (ng hr/mL) | 4,419.37 ± 335.65 | 5,113.23 ± 185.99[*] | 6,030.07 ± 793.69[*] |
| $AUMC_{0-24}$ (ng hr/mL) | 35,455.33 ± 1,775.56 | 41,606.67 ± 693.27[**] | 47,811.67 ± 2,889.41[**] |
| $MRT_{0-24}$ (hr) | 9.13 ± 0.21 | 9.44 ± 0.08[*] | 9.62 ± 0.29[*] |
| $MRT_{0-inf}$ (hr) | 12.21 ± 1.72 | 12.99 ± 1.04 | 14.34 ± 3.20 |

**Notes.**

Data is expressed as mean ± SD.

[#]5.6 mg/1.5 kg b.wt.

[a]Single dose treatment of aprepitant and single dose treatment of gliclazide.

[b]Multiple dose treatment of aprepitant (7 days) and single dose treatment of gliclazide.

Cmax, Maximum concentration.; Tmax, Time to maximum concentration; CL, Clearance; Vd, Volume of distribution; $T_{1/2}$, Half-life; AUC, Area under the curve; MRT, Mean residence time.

[*]$P < 0.05$ compared to gliclazide. group.

[**]$P < 0.001$ compared to gliclazide group.

intervals and insulin estimation at the time of peak concentrations) of gliclazide in normal and diabetic condition, (iii) it obtained results from two dissimilar species (rats and rabbits) for the precise identification, characterization and validation of the interaction, (iv) it determined the acute and chronic influence of aprepitant on gliclazide activity, (v) it determined the mechanism of pharmacokinetic interaction between aprepitant and gliclazide.

The rats are more sensitive to gliclazide effect compared to other animals and hence relevant dose selection is very crucial in drug–drug interaction studies. Hence, based on dose–response relationship study of gliclazide, the dose (2 mg/kg b wt) which can produce a blood glucose reduction of 30 to 35% was selected for the interaction studies in rats. The results from our dose–response relationship study are in compliance to the previous reports (*Satyanarayana & Kilari, 2006*; *Mastan & Kumar, 2009*; *Kumar & Mastan, 2011*).

Gliclazide produced a biphasic maximum response at 2 and 8 hr in rats which is probably due to an enterohepatic circulation (biliary excretion) of gliclazide as reported in previous animal studies and humans. The biphasic blood glucose reduction was not observed in rabbits confirming that there was no enterohepatic circulation (biliary excretion) of gliclazide in rabbits which is in consistent with the published studies (*Satyanarayana & Kilari, 2006*; *Mastan & Kumar, 2009*; *Kumar & Mastan, 2011*). In rats and rabbits, aprepitant had shown some marginal tendency to decrease the blood glucose and strong effects on insulin levels. The possible reason for this tendency is the inhibition of substance P release by aprepitant and its eventual relation with the decrease in blood glucose level (*Brown & Vale, 1976*; *Karagiannides et al., 2011*). In combination, the pharmacodynamic activity of gliclazide was significantly influenced by aprepitant in normal and diabetic condition. From these pharmacodynamic results, it appears to be an additive effect with the concomitant administration of aprepitant and gliclazide considering the individual contribution of aprepitant on blood glucose (modest effect) and insulin (strong effect) in rats and rabbits.

However, in pharmacokinetic study in rabbits, the gliclazide concentrations were significantly increased following the single- and multiple-dose treatment of aprepitant in rabbits. The increased concentrations and significantly altered pharmacokinetic parameters of gliclazide clearly indicate that there is a significant pharmacokinetic interaction between aprepitant and gliclazide. The increase in $C_{max}$ and AUC (bioavailability) of gliclazide in the presence of aprepitant suggesting that bioavailability of gliclazide was increased. The increase in bioavailability, increase in MRT and significant decrease in gliclazide clearance confirms the decrease in gliclazide metabolism by aprepitant. The significant inhibition of CYP3A4 by aprepitant with other drugs was reported in various studies (*Loos et al., 2007*; *Shindorf, Manahan & Geisler, 2013*; *Aapro & Walko, 2010*; *Majumdar et al., 2003*). In general, CYP3A4 inhibition is relatively higher with chronic administration of a drug and this phenomenon is reflected in our study (increase in concentrations, bioavailability and MRT, and, decrease in clearance of gliclazide). The possible mechanism of interaction is inhibition of CYP3A4 by aprepitant resulted in decrease in gliclazide metabolism and subsequently increases in gliclazide concentrations and bioavailability in rabbits. Overall, this study reasonably confirm that (a) the inhibition of CYP3A4 by aprepitant indeed lead to increased gliclazide concentrations and subsequently increased glucose reduction, (b) in

addition, the gliclazide pharmacodynamic activity was further influenced by the individual contribution of aprepitant on blood glucose (modest effect) and insulin (strong effect) in animal models.

Despite the highlights of this study, we admit that there are some limitations. Firstly, the pharmacokinetic study was not evaluated in diabetic condition. Second, there is no concomitant drug administration with gliclazide which is common in diabetes therapy. Hence, this study warrants further study to explore these parameters.

## CONCLUSION

In combination, aprepitant significantly influenced the pharmacodynamic activity of gliclazide in animal models. Considering this, care should be taken when this combination is prescribed for the clinical benefit in diabetic patients.

## ACKNOWLEDGEMENTS

The authors are thankful to M/s. Dr. Reddy's Laboratories, Hyderabad, for supplying gift samples of the experimental drugs for this study.

### Funding

The authors received no funding for this work.

### Competing Interests

Mastan Shaik is employed by Medical Services, Troikaa Pharmaceuticals Ltd., Ahmedabad, India.

### Author Contributions

- Raghunandan Reddy Kura conceived and designed the experiments, performed the experiments, analyzed the data, contributed reagents/materials/analysis tools, prepared figures and/or tables, authored or reviewed drafts of the paper, approved the final draft, statistical analysis and interpretation of the results.
- Eswar Kumar Kilari conceived and designed the experiments, analyzed the data, authored or reviewed drafts of the paper, approved the final draft.
- Mastan Shaik conceived and designed the experiments, contributed reagents/materials/-analysis tools, prepared figures and/or tables, authored or reviewed drafts of the paper, approved the final draft, statistical analysis and interpretation of the results.

### Field Study Permissions

The following information was supplied relating to field study approvals (i.e., approving body and any reference numbers):

The animal experiments were performed after prior approval of the study protocol by the Institutional Animal Ethics Committee and by the Government regulatory body for animal research (Reg. No. 516/01/A/CPCSEA). The study was conducted in accordance

with the guidelines provided by the Committee for the Purpose of Control and Supervision of Experiments on Animals (CPCSEA).

## Data Availability

The raw data are provided in the Supplemental File.

## Supplemental Information

Supplemental information for this article can be found online at http://dx.doi.org/10.7717/peerj.4798#supplemental-information.

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
