# Peer review of "Influence of aprepitant on the pharmacodynamics and pharmacokinetics of gliclazide in rats and rabbits"

_PeerJ, doi:10.7717/peerj.4798_

## Round 0.1 · original submission · Major Revisions

After you have completed your revisions, please provide a letter that details your modifications. If you disagree with the comments made by either reviewer, write a rebuttal and I shall adjudicate the matter.

Reviewer 1 ·

Basic reporting

Clear language has been used. Appropriate literature has been cited. The figures and tables have been professionally presented. The results section links well with the "hypothesis" of the paper.

Overall, the manuscript is written in good English and flows well. However, there are minor grammatical errors that can be easily rectified. For example, results section in the abstract- doe needs to be changed to dose. Results section- “these results confirming that …”, needs to be “these results confirm that …”. Discussion section, “The biphasic blood glucose reduction was note* observed in rabbits …”

In the introduction section, the line “In general, the overall … high risk individuals” seems to be out of place.

Experimental design

Original research, well defined research question, appropriate use of animal models and statistics

Validity of the findings

1. It is possible that the effect of concomitant administration of gliclazide and aprepitant on blood glucose level and insulin level (in normal rats, diabetic rats, and normal rabbits- tables 1, 2, and 3) is simply an additive effect, especially for the single dose treatment. Aprepitant alone seems to have a modest effect on blood glucose levels (and a strong effect on insulin levels- please show the data in the manuscript figures). Granted that the authors provide strong evidence suggesting the pharmacodynamics of gliclazide has been influenced (table 4), I would describe the effects of concomitant administration on insulin levels (figure 3) and serum gliclazide levels (figure 4) as modest changes. It would be optimal if the authors can convince the readers that these modest changes in insulin level and serum gliclazide levels indeed lead to increased glucose levels via pharmacodynamic interactions and not an additive effect.
2. The results section seems rushed. The authors must describe the results in detail using words rather than just referring to the tables and/or figures. It might also be useful to split the results section into smaller chunks using sub-headers such has pharmacodynamics, pharmacokinetics, etc.
3. In the results section, the authors say “These results confirming that the blood glucose reduction with gliclazide treatment was relatively higher in diabetic condition compared to normal condition”. Even though they have used the word ‘relatively’ in the sentence, it is not appropriate to make such comparisons without statistical evidence.

Reviewer 2 ·

Basic reporting

In this manuscript author tried to show the influence of aprepitant on pharmacodynamics (PD) and pharmacokinetics (PK) of gliclazide in rats and rabbits by analyzing glucose, insulin and drug bioavailability in blood samples of drug (gliclazide, gliclazide+ aprepitant) treated and non-treated rats and rabbits. It is a simple and straightforward study with valid reason and good idea (across species). Author also admitted couple of limitations, I appreciate that. However, I have couple of queries for this manuscript.

1. Author claimed/concluded that aprepitant significantly influenced the pharmacodynamic activity of gliclazide by CYP3A4 inhibition in animal models, while there was no data shown to the relevant claim. If author wants to claim aprepitant's inhibitory role on CYP3A4, either expressions levels of CYP3A4 or interaction studies of CYP3A4 with drug, correlating the PD and PK values should be shown.

2. Tables showing the values of normal and diabetic animals can be shown side by side or in the form of scatter plot. It would be much easier for reader to analyze.

3. Grammar/typing errors in line number 50 and 53

Experimental design

no comment

Validity of the findings

no comment

---

## Round 0.2 · accepted · Accept

No comments. Authors have comprehensively addressed both Reviewer's comments.

# Reviewer 1 ·

Basic reporting

The authors have revised the manuscript to overcome all grammatical and flow problems.

Experimental design

Look good.

Validity of the findings

The addition about the "additive effect" makes the manuscript scientifically sound.

Additional comments

The authors have addressed all of the reviewer comments.

Reviewer 2 ·

Basic reporting

Authors addressed all my concerns that i raised for this manuscript.
I recommend this manuscript for publication in peerJ journal.

Experimental design

no comment

Validity of the findings

no comment